# A chimeric viral platform for directed evolution in mammalian cells

Alexander J. Cole[1,3], Christopher E. Denes [2,3], Cesar L. Moreno[2], Lise Hunault[1], Thomas Dobson [1], Daniel Hesselson [1,4] ✉ & G. Gregory Neely [2,4] ✉

Directed evolution is a process of mutation and artificial selection to breed biomolecules with new or improved activity. Directed evolution platforms are primarily prokaryotic or yeast-based, and stable mammalian systems have been challenging to establish and apply. To this end, we develop PROTein Evolution Using Selection (PROTEUS), a platform that uses chimeric virus-like vesicles to enable extended mammalian directed evolution campaigns without loss of system integrity. This platform is stable and can generate sufficient diversity for directed evolution in mammalian systems. Using PROTEUS, we alter the doxycycline responsiveness of tetracycline-controlled transactivators, generating a more sensitive TetON-4G tool for gene regulation with mammalian-specific adaptations. PROTEUS is also compatible with intracellular nanobody evolution, and we use it to evolve a DNA damage-responsive anti-p53 nanobody. Overall, PROTEUS is an efficient and stable platform to direct evolution of biomolecules within mammalian cells.

Using iterative rounds of diversification, selection, and amplification, directed evolution (DE) can produce biomolecules with new or improved functions[1–6]. While this approach has been widely used to evolve molecules in simple prokaryotic and eukaryotic systems[7,8], these environments lack the full complement of post-translational modifications, protein-protein interactions, and signaling networks found in mammalian cells[9].

Ideally, proteins destined for mammalian applications would be evolved directly in mammalian cells. Historically, this has been achieved using ex mammalia mutagenesis techniques combined with phenotypic screening in mammalian cells[10]. More recently, targeted mutagenesis has linked protein function to selectable or screenable markers allowing target diversification and variant selection in the same mammalian cell[11–13]. However, cell-based approaches that link an integrated target molecule to cellular fitness can be derailed by mutations in the host genome[14]. Placing the target in a viral genome can mitigate this issue since naive host cells can be provided for each round of evolution. However, existing virus-based mammalian DE

systems are limited by safety concerns[15,16], low mutational rates[17], are target-specific[18,19], or lack functionality[20,21].

Here, we describe PROTein Evolution Using Selection (PROTEUS), a platform that uses chimeric virus-like vesicles (VLVs) to enable extended mammalian DE campaigns without loss of system integrity. PROTEUS rapidly generates authentic evolution products with superior functionality, and will have broad utility for evolving proteins designed to function in mammalian cells.

## Results

### Capsid-deficient VLVs support host-dependent propagation of the Semliki Forest Virus genome

Alphavirus genomic RNA is recognized by a strain-specific capsid protein that interacts with packaging signals that are distributed throughout the alphavirus genome[22]. In the context of DE, these interactions generate cheater particles that interfere with viral replication and contribute to a failure to recover authentic DE products[21]. While an intact capsid is essential for the pathogenicity of blood-borne

[1]Centenary Institute and Faculty of Medicine and Health, Charles Perkins Centre, The University of Sydney, Sydney, NSW, Australia. [2]The Dr. John and Anne Chong Lab for Functional Genomics, School of Life & Environmental Sciences, Charles Perkins Centre, The University of Sydney, Sydney, NSW, Australia. [3]These authors contributed equally: Alexander J. Cole, Christopher E. Denes. [4]These authors jointly supervised this work: Daniel Hesselson, G. Gregory Neely. ✉e-mail: d.hesselson@centenary.org.au; greg.neely@sydney.edu.au

viruses, the capsid protein is dispensable for in vitro propagation of VLVs[23].

To explore whether eliminating capsid-RNA interactions enables robust host-dependent viral propagation, we designed a chimeric two-component system based on a Semliki Forest Virus (SFV) replicon[24], which we modified to encode only non-structural viral proteins. In this system, the infectivity of SFV VLVs is determined by the expression

level of the Indiana vesiculovirus G (VSVG) coat protein from the host cell (BHK-21; Fig. 1a). Importantly, there are no regions of sequence homology between the RNA encoding VSVG and the SFV genome, reducing opportunities for recombination events that could restore replication competence. We generated an SFV replicon incorporating fourteen point mutations (ten non-synonymous, four synonymous) in the Non-Structural Proteins (NSPs 1-4), which were reported to

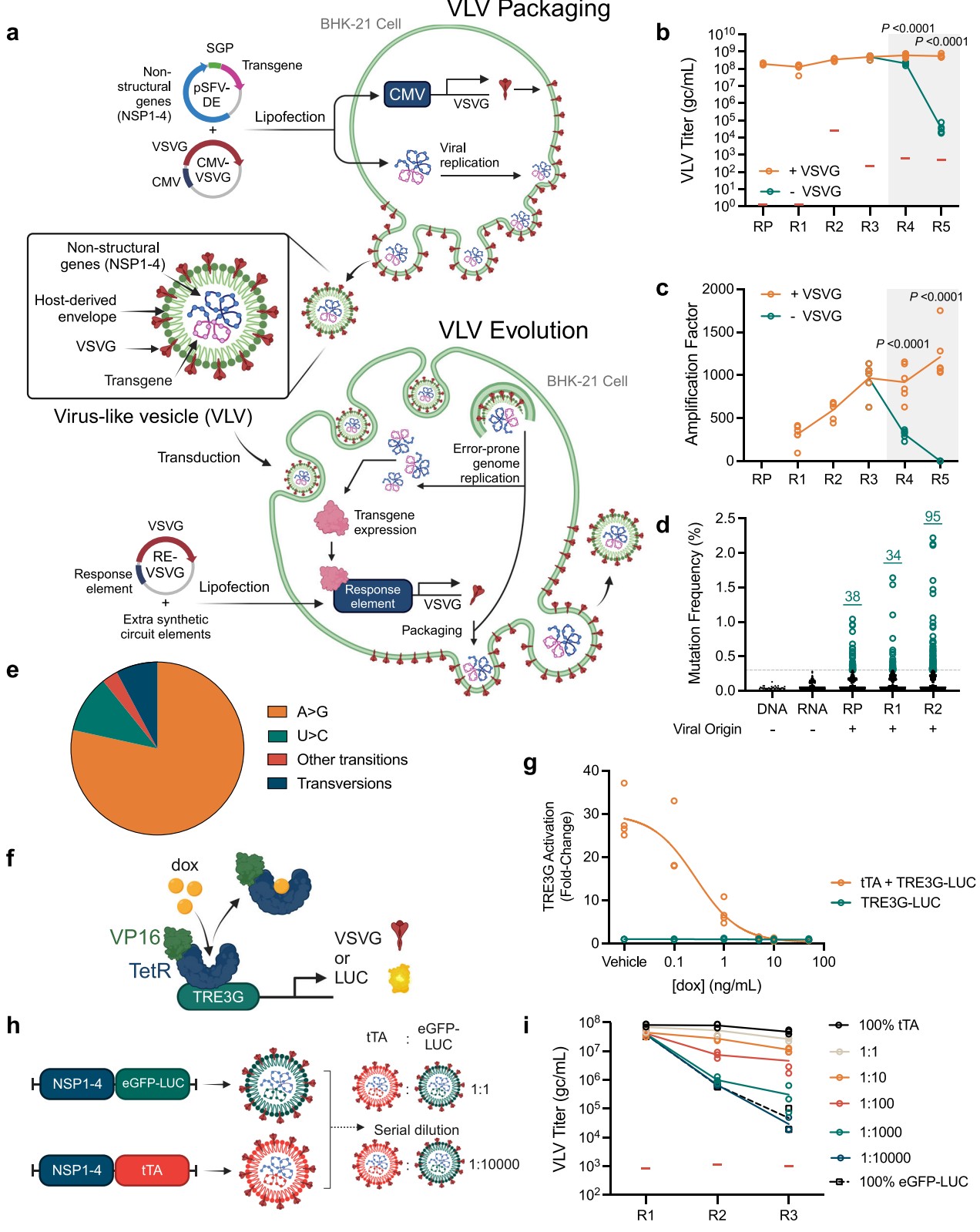

**Fig. 1 | Host-dependent propagation of VLVs. a** SFV VLVs are initially packaged using SFV-DE DNA replicons that encode the DE target transgene in cells that constitutively express VSVG. Infectious VLVs are propagated for evolution in host cells that constitutively express VSVG for VLV amplification or, for evolution, in cells expressing response element (RE)-controlled VSVG directly or indirectly regulated by the DE target. Titers (**b**) and amplification factors (**c**) of eGFP-LUC VLVs propagated in cells constitutively expressing CMV_VSVG ( + VSVG) for all rounds (R) from packaging (RP) to R5 or control DNA (-VSVG; for R4 and R5 presented within the gray box) ($N = 6$ biological replicates). Red bars indicate the RT-qPCR no template control (NTC) signal in (**b**). Statistical comparisons were made using two-tailed unpaired t-tests. **d** Allele frequency of mutations in a neutral eGFP-LUC transgene. Dotted line represents detection limit for viral variants (0.3%). **e** Mutational spectrum of viral variants from R2 in (**d**) ($N = 95$ variants detected). **f** Circuit design for tTA-mediated activation of TRE3G. **g** Dox-dependent repression of a TRE3G-regulated LUC reporter ($N = 4$ biological replicates). **h** Schematic of serial dilution assay. Colors are for illustrative purposes and do not reflect different membrane compositions. VLVs were transduced at MOI 1 at R1 and neat at R2 and R3. **i** Titers of serially diluted tTA VLVs propagated on cells expressing VSVG under the control of TRE3G in the absence of dox ($N = 3$ biological replicates). Red bars indicate the RT-qPCR NTC signal. Schematics for panels (**a**), (**f**) and (**h**) were created with BioRender.com. Source data are provided as a Source Data file.

increase the titer of SFV VLVs (Supplementary Fig. 1a[25],). Further, to reduce the cytopathic effects of SFV transduction, we exchanged a three amino acid loop within NSP2 with an attenuated variant (A674R/D675L/A676E[26],) to generate the pSFV-DE replicon construct (Fig. 1a, Supplementary Fig. 1a), named to indicate its application to directed evolution. Chimeric VLVs were produced by transfecting BHK-21 cells with the replicon vector and a vector constitutively expressing VSVG (pCMV_VSVG; Fig. 1a, VLV Packaging). Subsequent transduction of VLVs was performed in naive cells that were transfected to express VSVG (Fig. 1a, VLV Evolution). We found that attenuation of NSP2 did not affect VLV titer or amplification factor (the ratio of VLVs released per VLV transduced) (Supplementary Figs. 1b and c), indicating that reduced cytotoxicity was achieved without compromising VLV fitness.

To use chimeric VLVs for mammalian DE campaigns, their propagation must be dependent on the production of envelope protein VSVG by the host cell. To confirm host-dependence, VLVs carrying an eGFP-P2A-Luciferase (eGFP-LUC) transgene were propagated for multiple rounds at high titer (>$10^8$ genome copies (gc)/mL; Fig. 1b, R1-R3) and then used to infect VSVG- or mock-transfected host cells (Fig. 1b, R4-R5). VSVG-transfected cells supported VLV propagation, leading to an amplification factor >1000 (Fig. 1c). In the absence of VSVG, however, VLV titers fall rapidly and these mock-transfected (VSVG negative) cells exhibited an amplification factor <1.

While eGFP cargo expression was maintained in cells that constitutively expressed VSVG (Supplementary Fig. 1d), no eGFP expression was detectable after two rounds of transduction on mock-transfected cells (Supplementary Fig. 1e). Of note, VSVG-expressing cells showed a gradual reduction in eGFP levels over 5 rounds of transduction with VLVs (Supplementary Fig. 1d). Since VSVG levels were constitutive and independent of viral transgene activity, and since smaller viral genomes have a replicative advantage[27], we hypothesized that the reduced eGFP signal was due to truncation or loss of the eGFP-LUC transgene. Indeed, rounds of transduction were accompanied by progressive truncation of the viral transgene (Supplementary Fig. 1f), suggesting that selective pressure is required to maintain a full-length viral transgene.

DE requires diversification of the target transgene to produce variants with increased fitness. Alphaviruses carry error-prone RNA-dependent RNA polymerases with reported mutation frequencies >$10^{-4}$ per nucleotide in each round of replication[28]. Using non-viral DNA and RNA templates, we established a detection limit of 0.3% for new mutations by amplicon deep sequencing (Fig. 1d, dotted line). We observed an accumulation of bona fide mutations during the propagation of VLVs carrying the unselected eGFP-LUC cargo (Fig. 1d, RP-R2), and determined an overall mutation rate of 2.6 mutations/$10^5$ transduced cells. Multiple substitution types were observed with a strong A-to-G (and complementary U-to-C) transition mutational bias which is consistent with ADAR-dependent mutations (Fig. 1e, Supplementary Data 1[29]). Accordingly, ADAR/ADARB1 double knockout in the BHK-21 host cells eliminates ADAR activity (Supplementary Fig. 2a) and reduces the mutational bias (Supplementary Fig. 2b) but also decreases the overall mutation rate by 3-fold (from 2.6 to 0.8 mutations/$10^5$

transduced cells), impacting its utility for directed evolution. As such, wildtype BHK-21 cells were used for further applications.

The application of selective pressure to a target transgene requires a tight link between the expression level of VSVG and viral transgene activity (i.e., fitness). To test whether VLVs carrying a circuit-activating transgene gain a selective advantage, we first validated a simple circuit (Fig. 1f) where the tetracycline-controlled transactivator (tTA; TetOff, a fusion of TetR-VP16) activates a luciferase reporter under the control of an optimized tetracycline response element (TRE3G[30]) in the absence of doxycycline (dox) (Fig. 1g). Next, we independently packaged VLVs carrying either tTA or a circuit-neutral control eGFP-LUC transgene to enable competition testing. VLVs carrying the circuit-activating tTA transgene were serially diluted with VLVs carrying the neutral eGFP-LUC transgene and propagated in the absence of dox (Fig. 1h). At dilutions up to 1:1000, VLV populations containing the circuit-activating tTA transgene exhibited higher titers than the neutral eGFP-LUC control (Fig. 1i, and Supplementary Fig. 3). Thus, even rare tTA VLVs could outcompete dominant eGFP-LUC VLVs within 3 rounds. We next tested a serum-responsive circuit (Supplementary Fig. 4a and b[21],) in which expression of the serum response factor DNA binding domain (SRF[DBD]) fused to the VP64 activation domain (SRF-VP64) would stimulate circuit activation above background levels (Supplementary Fig. 4c). VLVs carrying the circuit-activating SRF-VP64 transgene exhibited a small proliferative advantage over those carrying control eGFP-LUC (Supplementary Figs. 4d and e). In this system, propagation of the neutral eGFP-LUC VLV population is likely supported by endogenous SRF activation of the introduced response element. The increased replication of the SRF-VP64 VLVs correlated with increasing prevalence of the transgene in a direct competition experiment within four rounds (Supplementary Fig. 4f–h). Selection rapidly favored a shorter SRF transgene that retained a minimal DNA-binding domain[31] (Supplementary Fig. 4i). Chimeric VLVs do not indiscriminately package significant amounts of VSVG RNA, and none was detected by R4 (Supplementary Fig. 4j), indicating that VLV propagation is entirely dependent on host cell complementary expression of VSVG. Overall, our chimeric system resolves problems with system integrity and cheater particle production identified in other alphavirus-based mammalian directed evolution systems[21].

## PROTEUS generates authentic evolution products

To test whether a circuit linking a VLV-encoded transgene to VSVG production can evolve protein function, we selected for dox resistance in tTA using the tTA-regulated synthetic circuit (Fig. 1f). Saturating clonal selections in bacteria have identified point mutations distributed across the tTA protein that confer dox resistance[32], some of which are also sufficient to provide dox resistance in mammalian cells[17]. Having confirmed that tTA activity was suppressed by dox in BHK-21 cells using a luciferase reporter under the control of TRE3G (Fig. 1g), we asked whether tTA activity could support VLV propagation on a TRE3G-regulated circuit under mild selective pressure. tTA and eGFP-LUC VLVs were independently packaged and amplified in cells constitutively expressing VSVG before switching to a TRE3G-regulated

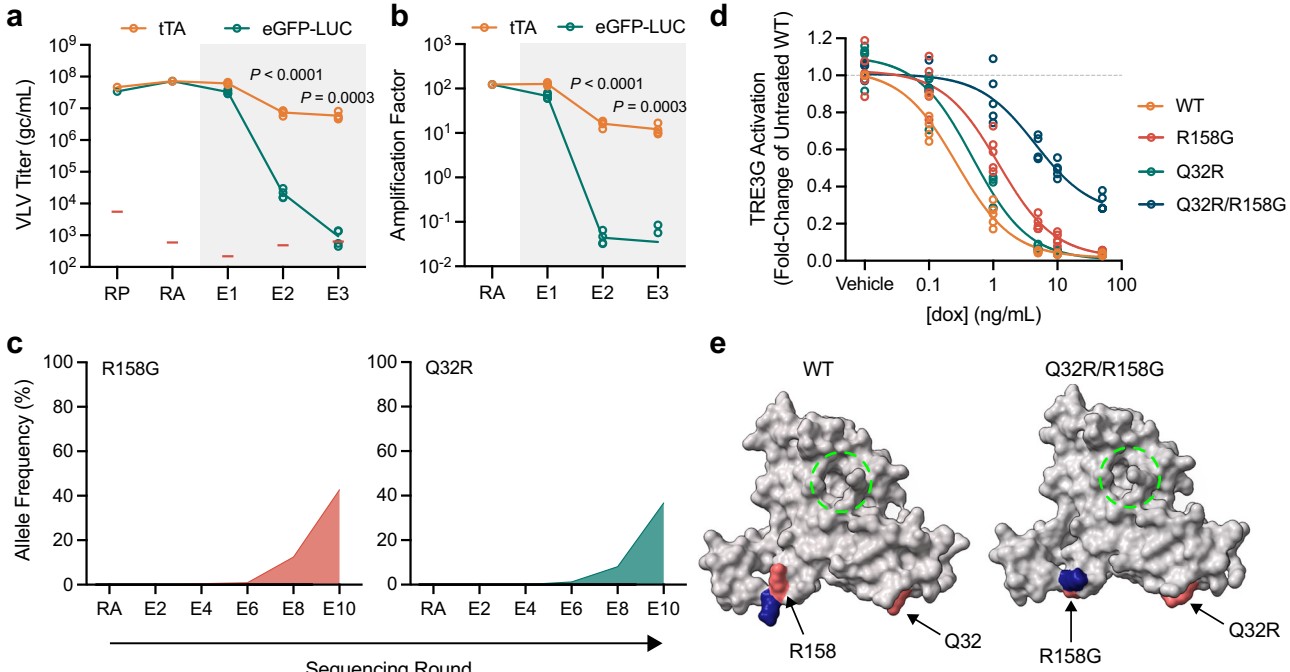

**Fig. 2 | PROTEUS generates authentic evolution products.** Titers (**a**) and (**b**) amplification factors of VLVs propagated on cells expressing VSVG under the control of TRE3G (gray box) (*N* = 4 biological replicates). RP (packaging) and RA (amplification) indicate rounds of VLV propagation under constitutive VSVG expression, while E1-X labeling indicates rounds of evolution under transgene-regulated VSVG expression. Red bars indicate the RT-qPCR NTC signal in (**a**). Statistical comparisons were made using two-tailed unpaired t-tests. **c** Allele frequency of the major variants identified in Campaign 1. **d** Dox-resistance of evolved tTA variants (*N* = 5 biological replicates). **e** Variant-induced structural changes in tTA modeled with AlphaFold2 (red, mutated residues; blue, displaced functional groups; dashed green circle, drug binding pocket). Source data are provided as a Source Data file.

circuit exposed to minimally inhibiting dox (Fig. 2a, b, 0.1 ng/mL, E1-E3, gray box), where tTA provided a large selective advantage over the neutral eGFP-LUC transgene. Note, all VLVs amplified in cells constitutively expressing VSVG show efficient transduction at E1 so the exponential selective advantage of circuit activation is observed from E2.

We conducted two independent ten-round evolution campaigns to isolate dox-resistant tTA variants (Supplementary Fig. 5a), which confirmed that, with appropriate transgene-responsive selective pressure, >96% of PROTEUS VLVs retain the transgene. An R158G variant (resulting from an A-to-G transition (Fig. 1e)) was detected in both campaigns by E4 (Supplementary Data 2), with different second-site mutations appearing by E6: Q32R (Campaign 1, Fig. 2c) and a triple mutant D178G/H179R/Q180R (Campaign 2, Supplementary Fig. 5b and c). Both variants from Campaign 1 have been previously identified as dox resistance mutants in *E. coli*[32,33]. In isolation, R158G provided modest dox resistance, while Q32R had minimal effect (Fig. 2d, Supplementary Data 3). However, the Q32R/R158G double mutant showed strong resistance to fully inhibitory concentrations of dox (Fig. 2d), consistent with their synchronized increase in frequency once both appeared in the population (Fig. 2c, Supplementary Data 2), although definitive co-occurrence cannot be evaluated from short-read sequencing data as these substitutions are too far apart. Similarly, in Campaign 2, the DHQ-to-GRR mutation enhanced the dox resistance of the initial R158G mutation (Supplementary Fig. 5d, Supplementary Data 3). Within this cluster of mutations at positions 178-180, all of the dox resistance was associated with the previously identified D178G variant (Supplementary Fig. 5e, Supplementary Data 3[32],), indicating that H179R/Q180R were passengers arising from a complex mutational event. Since structural modeling of the parental tTA protein with AlphaFold2[34] was congruent with the drug-bound crystal structure (Supplementary Fig. 5f), we modeled the dox resistant mutations from both campaigns, revealing side chain rearrangements in a linker region distal to the drug-binding

site (Fig. 2e, Supplementary Fig. 5g), highlighting the power of this system in generating non-obvious high fitness mutations. Together, these campaigns recovered authentic evolution products in tTA, validating chimeric VLVs as an efficient platform for directed evolution. We have named this platform PROTein Evolution Using Selection (PROTEUS).

## Enhancing drug-inducible transcriptional control

We next asked if PROTEUS can be used to improve existing molecular tools. Here, we focused on increasing the dox sensitivity of the third-generation reverse tetracycline-controlled transactivator (rtTA-3G; TetON), which has been extensively optimized using other methods[15,16]. The rtTA-3G transactivator is comprised of a bacterial reverse tetracycline repressor (rTetR) protein fused to a potent mammalian transcriptional activator (VP16). Using a TRE3G-regulated circuit (Fig. 3a), the parental rtTA-3G protein had an EC50 of 39 ng/mL dox (Fig. 3b). On the same circuit, rtTA-3G VLVs had a significant fitness advantage over neutral eGFP-LUC cargo VLVs at 100 ng/mL dox (Fig. 3c, Supplementary Fig. 6a). We propagated rtTA-3G VLVs for 30 rounds, adjusting the dox concentration to maintain strong selective pressure for increased dox sensitivity (Supplementary Fig. 6b). The VLVs were sequenced every 5 rounds (Supplementary Fig. 6c), revealing an M59I variant that appeared early in the campaign and reached fixation by E30 (Fig. 3d, Supplementary Data 4). A second D5N variant appeared by E20 and the majority of rtTA-3G transgenes (57.75%) carried both mutations by E30 (Fig. 3d, Supplementary Fig. 6d, Supplementary Data 4). No dominant additional variants were detected by E60 at 3 ng/mL dox suggesting that this campaign reached a local fitness peak. Both single mutations individually enhanced the dox sensitivity of rtTA-3G and the D5N/M59I double mutant was further improved (EC50 of 7 ng/mL; Fig. 3e, Supplementary Data 3), without increased leakiness in the absence of dox (Supplementary Fig. 6e).

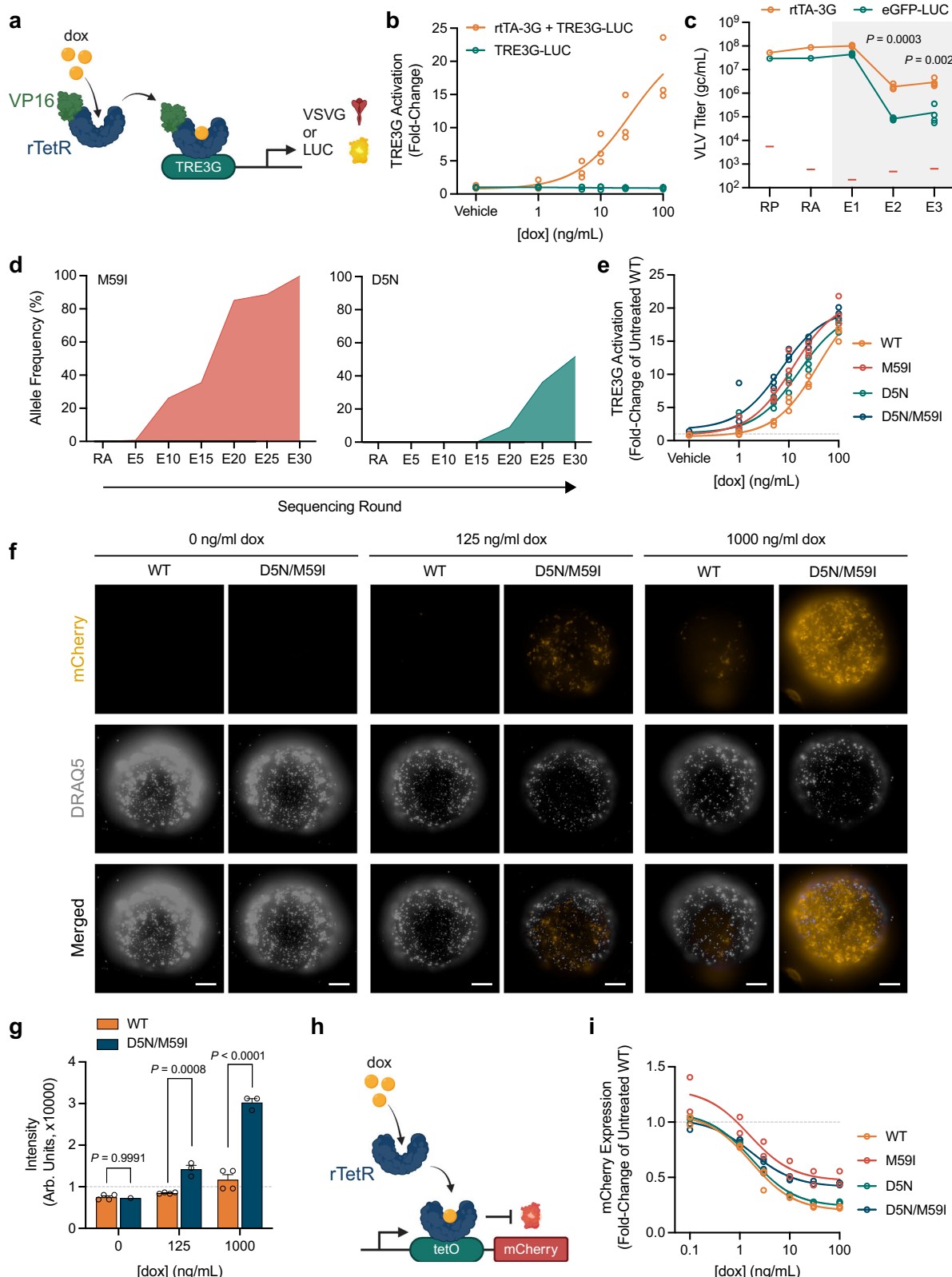

To evaluate if our evolved rtTA-3G gene induction system has broad utility, we tested its sensitivity using human iPSC-derived embryoid bodies. Embryoid bodies are 3D cell aggregates used for studying early human embryogenesis, but their non-vascularized state makes it difficult to deliver sufficient levels of dox to activate rtTA-3G and drive transgene expression[35]. Indeed, compared to the parental rtTA-3G, our DE product rtTA-3G[D5N/M59I] showed enhanced dox

sensitivity and mCherry reporter gene induction across a range of dox doses (Fig. 3f, g). To test whether D5N/M59I are mammalian-specific adaptations, we coexpressed rTetR in bacteria with an mCherry reporter that is repressed by rTetR in the presence of dox (Fig. 3h). In *E. coli*, the D5N/M59I single and double variants exhibited similar or lower sensitivity to dox when compared with the parental rTetR (Fig. 3i), suggesting that these mutations would not have been

**Fig. 3 | Enhancing drug-inducible transcriptional control. a** Circuit design for rtTA-mediated activation of TRE3G. **b** Dox-dependent activation of a TRE3G-regulated LUC reporter ($N = 3$ biological replicates). **c** Titers of VLVs propagated on cells expressing VSVG under the control of TRE3G at 100 ng/mL dox (gray box) ($N = 4$ biological replicates). Red bars indicate the RT-qPCR NTC signal. Statistical comparisons were made using two-tailed unpaired t-tests. **d** Allele frequency of the major variants identified during long-term propagation on minimal concentrations of dox. **e** Dox-sensitivity of evolved rtTA variants ($N = 3$ biological replicates). **f** Activation of evolved rtTA variants in embryoid bodies at DIV6; scale bar, 200 μm.

**g** Quantification of mCherry fluorescence in embryoid bodies presented in (**f**) (for D5N/M59I embryoid bodies, $N = 1$ biological replicate for 0 ng/mL dox, $N = 3$ biological replicates for 125 and 1000 ng/mL dox; $N = 4$ biological replicates for all WT embryoid bodies). Error bars represent mean ± SEM. Statistical comparisons were made using a two-way ANOVA with Šídák's multiple comparisons test to generate adjusted $P$ values. **h** Circuit design for rtTA-mediated inhibition of mCherry expression. **i** Dox-sensitivity of evolved rtTA variants in *E. coli* ($N = 2$-3 biological replicates). Schematics for panels (**a**) and (**h**) were created with BioRender.com. Source data are provided as a Source Data file.

obtained in a bacterial DE system. Both mutations were distal to the dox binding site and were not predicted to cause any major structural changes (Supplementary Fig. 6f). Overall, the D5N/M59I variant has a superior and mammalian-specific dox response profile and may be useful as a fourth-generation rtTA tool (rtTA-4G).

### Directed evolution of an intracellular nanobody

Intracellular nanobodies (Nb) have potential for interrogating or modulating protein function, though in some cases further optimization may be required[36]. For example, a p53 biosensor based on the p53-interacting nanobody Nb139 (Supplementary Fig. 7a[37],) does not localize to the nucleus in response to cisplatin (Supplementary Fig. 7b) despite robust nuclear p53 accumulation (Supplementary Fig. 7c). To improve the intracellular function of Nb139, we conducted a DE campaign using a 2-hybrid circuit design with a p53 bait (Fig. 4a). The parental nuclear localization signal (NLS)-containing Nb139-VP64 fusion successfully activated a reporter circuit (Fig. 4b, Supplementary Fig. 7a), and Nb139-VP64 VLVs outcompeted neutral eGFP-LUC cargo VLVs (Fig. 4c, Supplementary Fig. 7d). Activation of this reporter measures the integrated effects of Nb139 stability, affinity for p53, and nuclear localization. Long-term propagation of Nb139-VP64 VLVs (Supplementary Fig. 7e) led to the accumulation of S26P and Y60C mutations that evolved to fixation in the population (Fig. 4d, Supplementary Fig. 7f, Supplementary Data 5). S26P and Y60C map to framework region (FR) 1 and FR2, respectively, indicating that these variant residues do not interact directly with p53 (Fig. 4e). However, only the S26P variant increased reporter activity at early and late timepoints (Fig. 4f), with no further increase observed in the S26P/Y60C double mutant.

To evaluate the Nb139 variants as biosensors to monitor p53 activity, we generated versions of parental and single or double S26P or Y60C mutant Nb139 that no longer contained an NLS and replaced the VP64 domain with an eGFP reporter (Supplementary Fig. 7a), and then observed cytosolic to nuclear localization following DNA damage. In response to cisplatin treatment, Nb139[S26P]-eGFP translocated to the nucleus and labeled nuclear puncta prior to cell death, in contrast to more uniform expression throughout the cell in turboGFP and Nb139[WT] controls (Fig. 4g, Supplementary Fig. 8, Supplementary Movies 1–3). The S26P single mutant exhibited the largest response to cisplatin, while Y60C also modestly improved sensitivity in the biosensor format (Fig. 4h). This campaign demonstrates that the intracellular function of Nb139 (crystal structure 4QO1), which was already classified as stable[36], can be further improved through evolution within a mammalian cell. Notably, the evolved variants did not directly alter the Nb139-p53 binding interface. Overall, our application of PROTEUS here has generated an evolved p53 biosensor that allows the visualization of p53 nuclear recruitment in vitro.

### Broadening accessible evolutionary sequence space

PROTEUS-driven directed evolution results primarily in ADAR-dependent A-to-G mutational bias (Fig. 1e). To expand the genetic diversity of PROTEUS directed evolution[38], we tested multiple antiviral drugs that interfere with viral RNA polymerases[13,39]. At a sub-lethal dose (Supplementary Figs. 9a and b), we found the RNA nucleoside analog molnupiravir increased the PROTEUS mutation rate 8-fold (to 22

mutations/$10^5$ cells) and redistributed mutational bias as has recently been reported by others[13,39] (Supplementary Figs. 9c and d). Overall inclusion of molnupiravir or similar mutagens may enhance the utility of PROTEUS for directed evolution campaigns in the future.

## Discussion

Mammalian directed evolution aims to generate biomolecules that are optimized in situ for a desired activity. This is dependent on both a high mutation rate to diversify the target and a robust link between function and fitness to maintain the integrity of the system, technical challenges that have not yet been overcome. To this end, we have developed PROTEUS, a directed evolution platform based on a chimeric SFV design, which takes advantage of the high error rate inherent in alphavirus replication while solving major issues with previous strategies[21].

Natural viral infections generate a diversity of virions encoding incomplete genomes that collectively compete with host defenses[40]. Our work was motivated by the recent observation that capsid-genome interactions are distributed throughout the alphavirus genome[22], which contribute to the production of non-functional cheater particles containing incomplete genomes that encode structural proteins and contaminate evolution campaigns[21]. The capsidless VLVs used in PROTEUS utilize the cellular exosome pathway to escape the host cell[25], eliminating the co-evolved and highly redundant links between the viral genome and its cognate capsid. Our chimeric viral system maintained system integrity and enabled long-term propagation (>30 rounds) on a synthetic evolution circuit. It is not yet clear how many rounds of PROTEUS evolution are required to maximize mutational space; these are likely circuit- and target-specific parameters.

We confirmed that PROTEUS generates authentic evolution products by recovering well-characterized dox resistance mutations in tTA. While single mutations appeared in the VLV population at low frequency, they were rapidly outcompeted by double mutant variants that exhibited greater dox resistance. These mutant combinations highlight the power of PROTEUS to generate complex mutations that would be difficult to generate through existing technologies. For example, sampling the complete double mutant sequence space of tTA would require testing >$10^6$ combinations, which is beyond any existing experimental approach. In a more challenging test, the sensitivity of the highly optimized rtTA-3G protein was further increased by PROTEUS using long-term propagation on limiting amounts of dox. Importantly, our enhanced rtTA-3G protein is a mammalian-specific adaptation that does not show similar increased sensitivity in bacteria. Finally, PROTEUS was useful in optimizing a p53 nanobody into a biosensor that can detect a DNA damage response within a mammalian nucleus.

Our study does have limitations. The mutational bias we observed is consistent with ADAR-mediated editing of the SFV RNA genome[41]. Although knockout of ADAR activity in host cells reduces the mutation spectrum bias, it also lowers the mutation rate. We show that treating transduced cells with RNA nucleoside analogs such as molnupiravir can increase the mutation rate and change the mutation spectrum, however, the basal alphavirus mutation rate was sufficient for the evolution campaigns presented here. Additionally, VSVG expression

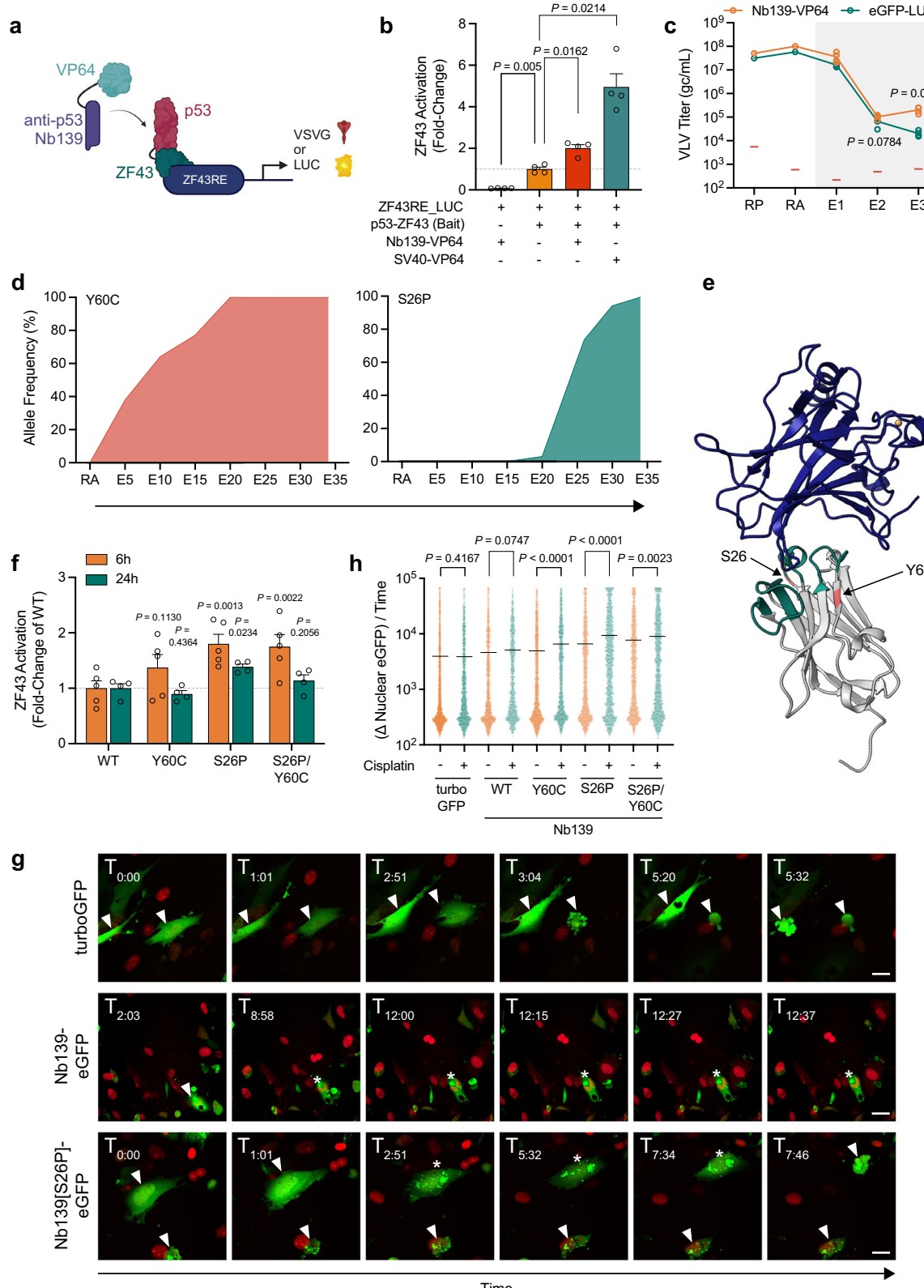

must exceed a certain threshold to support VLV propagation. The circuit activity can be precisely tuned when selective pressure is controlled by a small molecule (e.g., dox), however, it may be more challenging to maintain this balance for more complex targets (e.g., receptors, ligands, signaling molecules) or with more complex circuit designs. Finally, while improved prokaryotic directed evolution systems enable shorter campaigns with very high mutation rates[42],

PROTEUS provides unique utility for targets where a mammalian cellular context is critical. PROTEUS currently functions in hamster BHK-21 cells which are routinely used for alphavirus propagation because they lack a functional antiviral Type I interferon response[43]. Targeted inhibition of this pathway could pave the way for the adaptation of PROTEUS to other mammalian cell types to provide tissue- or disease-specific environments for directed evolution campaigns.

**Fig. 4 | Directed evolution of an intracellular nanobody. a** 2-hybrid circuit design for nanobody-p53 interactions. Schematic was created with BioRender.com. (**b**) Binding of Nb139-VP64 to a p53 bait activates the 2-hybrid circuit, measured using a LUC reporter (SV40, positive interaction control[53]; $N = 4$ biological replicates)). Data is normalized to the ZF43RE_LUC + p53-ZF43 bait incomplete circuit (orange). Error bars represent mean ± SEM. Statistical comparisons were made using a repeated measures one-way ANOVA with the Geisser-Greenhouse correction with a Dunnett's multiple comparisons test to generate adjusted *P* values. **c** Titers of VLVs propagated on cells expressing VSVG under the control of a p53 2-hybrid circuit (gray box) ($N = 4$ biological replicates). Red bars indicate the RT-qPCR NTC signal. Statistical comparisons were made using two-tailed unpaired t-tests. **d** Allele frequency of the major variants identified during long-term propagation on the 2-hybrid circuit. **e** Crystal structure 4QO1 showing Nb139 (gray; green, Nb139

complementarity-determining regions; red, evolved variant positions) in a complex with p53 (blue). **f** Evaluation of evolved Nb139-VP64 variants using the p53 bait reporter assay from (**b**) ($N = 4$ biological replicates). Error bars represent mean ± SEM. Statistical comparisons were made using a repeated measures one-way ANOVA with a Dunnett's multiple comparisons test to generate adjusted *P* values. **g** Timelapse of cisplatin-treated cells that express parental Nb139-eGFP, Nb139[S26P]-eGFP fusion or turboGFP alone (red, nuclei labeled with mCherry; scale bar, 25 µm). White arrows indicate cells of interest; asterisks indicate foci formation. Representative of $N = 2$ biological replicates. **h** Quantification of changes in nuclear eGFP expression over time from timelapse data represented in (**g**) ($N > 1000$ tracked cells). Error bars represent population means. Statistical comparisons were made using one-tailed unpaired t-tests. Source data are provided as a Source Data file.

Overall, we show that PROTEUS can be used to evolve protein activities within the context of a mammalian cell. We anticipate that PROTEUS will help the research community generate or optimize diverse biomolecules designed to function in complex mammalian systems.

## Methods

### Cell culture
BHK-21 [C-13] cells (#CCL-10) were sourced from the American Type Culture Collection (ATCC). Wildtype and CRISPR/Cas9-modified BHK-21 cells were maintained in a humidified 37 °C (5% $CO_2$) atmosphere in MEM α (ThermoFisher, #32571101) supplemented with 5% fetal bovine serum (FBS) (Bovogen, #SFBS-F, French origin) and 10% tryptose phosphate broth (TPB) (ThermoFisher, #CM0283B), hereafter defined as 'BHK-21 Growth Medium'. Fresh aliquots of cells were thawed for directed evolution campaigns once in-house passage counts reached 28.

HEK293T cells (female) were sourced from the ATCC (CRL-3216). Cells were maintained in a humidified 37 °C (5% $CO_2$) atmosphere in Dulbecco's Modified Eagle Medium (ThermoFisher, #11995065) supplemented with 10% Fetal Bovine Serum (Bovogen, #SFBS-F) and 100 U/mL penicillin and 100 µg/mL streptomycin (ThermoFisher, #15140122).

### Molecular biology and plasmid construction
All PROTEUS system and synthetic circuit plasmids were designed in SnapGene® (Version 6.1). Sequences of interest (e.g., promoter sequences, transgene inserts, plasmid vector backbones etc.) were isolated by high-fidelity PCR amplification with Velocity DNA Polymerase (Bioline, #BIO-21099) or Q5® High-Fidelity 2X Mastermix (NEB, #M0492) using primers synthesized by Integrated DNA Technologies (IDT). Assembly of PCR amplicons was performed using the NEBuilder HiFi DNA Assembly Master Mix (NEB, #E2621) following the manufacturer's recommendations. Assembled products were transformed into NEB® 10-beta Competent *E. coli* (NEB, #C3019) and selected on LB agar plates (ThermoFisher, #22700025) supplemented with 100 µg/ml ampicillin (Sigma-Aldrich, #A9518). Single colonies grown overnight in liquid LB broth (ThermoFisher, #12795-084) supplemented with 100 µg/ml ampicillin were processed with the ISOLATE II Plasmid Mini Kit (Bioline, #BIO-52057) for sequence verification. Plasmid constructs were verified by restriction digestion and Sanger sequencing at the Australian Genome Research Facility (AGRF). Once successful assembly had been confirmed, fresh streaks of confirmed positive constructs were generated on antibiotic-supplemented LB agar plates and cultures generated for plasmid isolation using the PureYield™ Plasmid Maxiprep System (Promega, #A2393), producing the yields and purity necessary for transfection applications. Plasmids used or generated in this study are listed in Supplementary Data 6.

**PROTEUS Virus-Like Vesicle (VLV) SFV constructs.** The initial SFV DNA plasmid construct (pSFV-DE_eGFP-LUC) was designed for expression from a constitutive CMV promoter in the mammalian expression vector pcDNA3.1(+). This parental plasmid was linearized by PCR and the amplicon purified by gel extraction. SFV elements

(5'UTR, NSP1-4, 3'UTR), the sequences of which were derived from pSFV3 (Addgene #92072)[44], were synthesized as a series of gBlocks HiFi Gene Fragments by IDT and used for NEBuilder HiFi assembly. We built fourteen point mutations into the NSP genome sequence, which evolved in a capsid-deficient SFV strain for production of high-titer VLVs enveloped with VSVG[25]. Two intermediates containing separate halves of the NSP1-4_eGFP-LUC_3'UTR insert were generated from which the final assembly was obtained by PCR and HiFi assembly.

Subsequently, the NSP2 $^{674}$ADA$^{676}$ codon Vloop was mutated to $^{674}$RLE$^{676}$ by HiFi assembly to attenuate cytopathic effects[45]. This attenuated VLV (pSFV-DE) forms the basis of the SFV DNA replicon system presented within this study, and all transgenes for directed evolution were inserted as a direct replacement of the eGFP-LUC coding sequence by PCR and HiFi assembly.

### Packaging (RP) of PROTEUS VLVs
PROTEUS VLVs were packaged in BHK-21 cells following transfection with plasmid DNA ('VLV Packaging', Fig. 1a). BHK-21 cells were seeded in 6-well plates at a density of $1.95 \times 10^5$ cells/well in 2 mL BHK-21 Growth Medium and incubated for 24 h. A total of 1 µg of plasmid DNA (1:1 of pCMV_VSVG, pSFV-DE_[transgene]) was diluted in Opti-MEM Reduced Serum Medium with GlutaMAX Supplement (ThermoFisher, #51985034) and transfected into cells using *Trans*IT-2020 Transfection Reagent (Mirus Bio, #MIR5400) following the manufacturer's recommendations (employing a 3:1 transfection reagent:DNA ratio). 6 h post-transfection, cells were rinsed twice with DPBS (Sigma-Aldrich, #D8537) and 1.2 mL of BHK-21 Growth Medium was added. VLV-containing supernatant was collected 24 h post-transfection and centrifuged at 1000 g for 5 minutes to pellet cellular debris. Clarified supernatants were collected for titration and subsequent transduction experiments. VLVs produced during packaging are referred to as Round P (RP) samples.

For evolution campaigns, pSFV-DE_[transgene] replicon constructs were co-packaged with the circuit-neutral pSFV-DE_eGFP-LUC (plasmids were transfected in a 2:1:1 ratio of pCMV_VSVG, pSFV-DE_[transgene], pSFV-DE_eGFP-LUC as above) to enable visualization of transduction success through eGFP expression in early rounds, and to confirm transgene enrichment under selective pressure over rounds by sequencing.

### VLV titration
VLV-containing supernatants were titrated as per[21] using an NSP2-specific primer-probe set (Supplementary Data 7). Briefly, following collection and clarification, undiluted VLV-containing supernatants were combined with the TaqMan™ Fast Virus 1-Step Master Mix (ThermoFisher, #4444434) in a minimum of technical duplicates. Plates were run on a QuantStudio™ 7 Flex or QuantStudio™ 6 Pro Real-Time PCR System (ThermoFisher) using 'Fast' protocol parameters. Serially diluted pSFV_eGFP-LUC plasmid DNA was used to generate a standard curve for absolute quantification (ranging between $10^2$–$10^7$ genome copies (gc) per reaction), diluted in BHK-21 Growth Medium. Plotted standard curves were used to determine VLV titers in gc/mL.

The round-specific no-template control (NTC) signal is indicated with red bars on VLV titration plots, calculated from reactions using BHK-21 Growth Medium instead of VLV-containing supernatant. All titer plots portray unadjusted titer values, though NTC signals were subtracted when determining calculations for subsequent transductions. For tTA, rtTA-3G, and Nb139-VP64 campaigns, the presented RP-E3 rounds were performed in parallel and therefore share RT-qPCR NTC signals.

## VLV transduction (Amplification (RA) and Evolution round 1 (E1)-onwards)

**Amplification (RA).** Packaged VLVs from RP were propagated further in constitutive VSVG-expressing BHK-21 cells to amplify infectious, enveloped particles ('VLV Evolution', Fig. 1a). BHK-21 cells were seeded in 6-well plates (1.95 x 10^5 cells/well in 2 mL) in BHK-21 Growth Medium and incubated for 24 h. A total of 1 µg of pCMV_VSVG DNA was transfected into cells using *Trans*IT-2020 Transfection Reagent. 6 h post-transfection, cells were rinsed twice with DPBS and neat RP VLVs added (500 µL for 6-well plates) in the presence of 8 µg/ml polybrene (Sigma-Aldrich, #H9268). Cells were incubated with VLVs for 1 h and rinsed twice with DPBS before 1.2 mL BHK-21 Growth Medium was added for a further 23 h of incubation. VLV-containing supernatants were collected and processed as described in *Packaging (RP)*. VLVs produced during amplification are referred to as Round A (RA) samples.

**Evolution (E1 and onwards).** For evolution, amplified VLVs from RA were propagated in synthetic circuit-expressing cells that require VLV-encoded transgene functionality to activate transcription of the VSVG packaging element ('VLV Evolution', Fig. 1a). BHK-21 cells were seeded in 6-well plates at a density of $1.95 \times 10^5$ cells/well in 2 mL BHK-21 Growth Medium, incubated for 24 h, and transfected using *Trans*IT-2020 Transfection Reagent (Mirus Bio, #MIR5400) following the manufacturer's recommendations with a total of 1 µg of synthetic circuit-encoding plasmids. The SRF circuit used 1 µg of pSRE_VSVG; the tTA/rtTA-3G circuits used 1 µg of pTRE3G_VSVG; the Nb139 circuit supplied two plasmids at a 1:1 ratio (p53-ZF43:pZF43-VSVG), 0.5 µg/plasmid. Growth medium was replaced prior to transfection if the circuit of interest required chemical additives or adjusted media (e.g., reduced [FBS]). 6 h post-transfection, a mock-transfected well was trypsinized and counted to calculate the volume of titered VLV inoculum needed to achieve an MOI of 1 gc/cell. Typical counts ranged between $3-10 \times 10^5$ cells/well. An MOI of 1 is used to restrict circuit activation to the activity of a single transgene variant per cell to minimize crosstalk. Cells were rinsed once with DPBS before VLVs were applied in a 500 µL volume of BHK-21 Growth Medium supplemented with 8 µg/ml polybrene. Cells were incubated with VLVs for 1 h and rinsed twice with DPBS before 1.2 mL BHK-21 Growth Medium was added for a further 23 h of incubation (supplemented with chemical additives (e.g., dox or reduced [FBS] where relevant)). VLV-containing supernatants were collected and processed as described in *Packaging (RP)*. For transactivator evolution campaigns, the concentration of dox used to initiate these experiments was optimized to permit propagation of VLVs.

For all circuits tested, each subsequent round uses the titered VLVs produced in the preceding round to iteratively diversify, select, and amplify variants of improved fitness. The round-specific RT-qPCR NTC signals were subtracted from VLV titers prior to MOI calculations.

Note that the number of cells transduced, and hence the number of variant transgene-carrying VLVs screened per round, can be increased by proportionally upscaling to larger flask footprints. All evolution campaigns presented within this article represent experiments performed in 6-well plates. An MOI > 1 could be used to achieve higher circuit activation, but dominance of high fitness mutations may be delayed by the piggybacking of low fitness variants following co-transduction of a single cell.

For mutation bias experiments, BHK-21 Growth Medium was supplemented with molnupiravir (Sapphire Bioscience, #29586;

reconstituted in 100% DMSO) during the transfection, transduction, and replication phases of the round.

## VLV mixing assay

To determine the sensitivity of PROTEUS, tTA and eGFP-LUC VLVs were packaged for a mixing experiment. As per *Evolution (E1 and onwards)* above, at R1, these VLVs were transduced into BHK-21 cells expressing VSVG under the tTA-responsive TRE3G response element at an MOI of 1 gc/cell, to represent 100% tTA and 100% eGFP-LUC VLV populations. Simultaneously, for mixed populations, an MOI of 1 was maintained for eGFP-LUC and this was combined with serially diluted tTA VLVs (diluted from an MOI of 1 by 1:10 - 1:10000). For R2 and R3, cells were transduced with undiluted VLVs from the previous round, rather than at a specific MOI. VLV-containing supernatants were collected and processed as described in *Packaging (RP)*.

## Generation of CRISPR knockout cell lines

ADAR and ADARB1 single guide RNAs (sgRNAs; Supplementary Data 6) were ordered from Synthego and reverse transfected into BHK-21 cells according to the manufacturer's protocol. 72 h post-transfection, genomic DNA was isolated with the DNA Blood & Tissue Kit (Qiagen, #69504) and complementary target regions amplified and Sanger-sequenced. Inference of CRISPR Edits (ICE, https://ice.editco.bio/#/[46],) was used to determine editing efficiency. For guides with > 30% editing efficiency, single cell clones were screened to establish KO cell lines. Low-efficiency guides were incorporated into plentiCRISPRv2_puro (Addgene, #98290) at BsmBI (NEB, #R0739) sites to allow for antibiotic selection. plentiCRISPRv2_puro carrying target sgRNAs were transfected into HEK293T cells using Lipofectamine 3000 (Invitrogen, #L3000015) along with third generation gag, pol and rev lentivirus packaging plasmids. Medium was replaced after 16 h and lentivirus-containing supernatant collected 24 h later. Lentivirus was then used to transduce BHK-21 cells under selection with puromycin as previously described[47]. Editing efficiency was determined as described above. To validate ADAR and ADARB1 activity, ADAR reporter constructs were designed by inserting a dsRNA hairpin containing a stop codon[48] upstream of a luciferase reporter (Supplementary Data 6). Reporter constructs were transfected into ADAR/ADARB1 KO cell lines.

## Luciferase/resazurin assay

BHK-21 cells were seeded in $2 \times 96$-well plates at $6.6 \times 10^3$ cells/well. Cells were transfected with a total 34 ng of plasmid constructs using *Trans*IT-2020 Transfection Reagent following the manufacturer's recommendations. At 6 h post-transfection, cells were washed twice with DPBS, fresh BHK-21 Growth Medium added and the plates incubated for 24 h. Cell viability (as a proxy for cell density) was determined by replacement of growth medium with 30 µg/ml resazurin (Sigma-Aldrich, #R7017)-supplemented medium and incubation for -30 min. Resorufin fluorescence was measured on an Infinite M1000 PRO microplate reader (Tecan). Supplemented medium was removed before cells were rinsed once with room temperature DPBS and replaced with 30 µL growth medium. Luciferase activity was assessed using the Steady-Glo® Luciferase Assay System (Promega, E2510) following the manufacturer's recommendations in black-bottom plates. Firefly luminescence was measured on an Infinite M1000 PRO microplate reader (Tecan). Raw firefly luminescence values were background-subtracted and then normalized against well-matched resazurin fluorescence prior to analyzes.

## Transgene isolation

SFV RNA was extracted from 140 µL culture supernatant using the QIAamp Viral RNA Mini Kit (Qiagen, #52904) following the manufacturer's recommendations. SFV RNA was reverse transcribed into cDNA and all SFV transgene sequences, positioned between NSP4 and the viral 3' untranslated region (UTR), were PCR-amplified using the

OneTaq® One-Step RT-PCR Kit (E5315S). PCR was conducted using an NSP4-specific forward primer and a viral 3'UTR-specific reverse primer (listed in Supplementary Data 7). Amplified transgenes were resolved by gel electrophoresis using standard protocols and amplicons of the appropriate size range for the transgene of interest were purified by gel extraction using the ISOLATE II PCR and Gel Kit (Bioline, #BIO-52060). Amplicons were sequenced through long-read nanopore, short-read Illumina, or single read Sanger sequencing approaches. For Sanger sequencing, amplicon transgene DNA was blunt cloned into the linear plasmid backbone of the PCR Cloning Kit (NEB, #E1202S) using manufacturer's recommendations. Following transformation, single colonies were cultured and sequenced as described in *Molecular Biology and Plasmid Construction*.

### Long-read (Oxford nanopore) sequencing
**Sample processing.** RT-PCR-isolated transgene DNA sequences were processed to generate the libraries required for full-length transgene sequencing with Oxford Nanopore Technologies (ONT) Flongle flow cells (ONT, #FLO-FLG001). Sequencing samples were prepared following the recommendations found within the ONT protocol 'Amplicons by Ligation' (version ACDE_9110_v110_revV_10Nov2020). DNA concentrations were determined using the Qubit™ dsDNA HS Assay Kit (ThermoFisher, #Q32851) and 200 fmol of DNA was processed using the NEBNext Companion Module for ONT Ligation Sequencing (NEB, #E7180). Sequencing adapters were ligated onto sequences with the Ligation Sequencing Kit (ONT, #SQK-LSK110).

**Sequencing and basecalling.** Up to 40 fmol of each DNA library was loaded on ONT Flongle flow cells (R9.4.1 chemistry, #FLO-FLG001) in a MinION Mk1B Sequencer (ONT, #MIN-101B) fitted with a Flongle Adapter (ONT, #ADP-FLG001). Sequencing was performed using MinKNOW (ONT, version 4.2.8) using default parameters and the following inputs: kit used, SQK-LSK110; 0.5 h between MUX scans; basecalling, disabled. A minimum $2 \times 10^4$ raw reads were obtained per flow cell. Raw FAST5 files were basecalled using Guppy (version 4.5.2) with the minimum q-score filter set to 7.0. Basecalled FASTQ reads have been deposited at the Gene Expression Omnibus (GEO; GSE250502).

**Alignment.** Quality-filtered basecalled reads (in FASTQ format) were processed using EPI2ME Desktop Agent (ONT, version 3.3.0.1031). Sequence reference files were uploaded into the program with the Fasta Reference Upload workflow (v2021.07.15). For each sample, all reads were aligned to reference files using the Fastq Custom Alignment workflow (v2021.03.25) using default parameters.

### Short-read (Illumina) sequencing
**Sample processing.** RT-PCR-isolated transgene DNA sequences were processed further to generate the yield of DNA required for short-read Illumina sequencing with NovogeneAIT Genomics Singapore (Novogene). Extracted DNA was further amplified by high-fidelity PCR amplification with Q5® High-Fidelity 2X Mastermix (NEB, #M0492) using the same primer set as for initial transgene isolation. Amplicons were resolved by DNA gel electrophoresis and purified to have a minimum 1.5 µg for sequencing.

**Sequencing.** Samples were sequenced with Novogene using their 'Microbial PCR Product Whole Genome Sequencing' strategy. Amplicons underwent fragmentation prior to PCR-free library preparation and sequencing using NovaSeq PE250 technology. Basecalled FASTQ reads have been deposited at the Gene Expression Omnibus (GEO; GSE250502).

**Variant nucleotide analysis.** To identify variant nucleotides, basecalled FASTQ files were aligned to transgene reference sequences (FASTA) with *minimap2* (v2.24). Any mate-pair issues were fixed in the output .sam files with the *samtools* (v1.16) *fixmate* command before reads were sorted (*samtools sort*) and reformatted (*samtools mpileup*). VarScan 2.3.9[49] was used to call for variants with command *mpileup2snp* with arguments *--min-coverage* 10000 *--min-reads2 15 --min-var-freq* 0.00025 *--p-value 0.01*. Further annotation was performed with *vcf-annotator* (v0.7) using default commands, aligning to transgene reference sequences (GenBank format).

### Immunocytochemistry
Cells were fixed with 4% PFA for 20 minutes at room temperature. Blocking was performed in PBS with 5% Normal Goat Serum, 1% BSA, 0.05% Triton-X, 0.3M Glycine for 1 h at room temperature. Cells were then incubated with 1:400 anti-P53 mAb (Clone 7F5) (Cell Signaling Technology, #2527) in blocking solution for one hour at room temperature, followed by five washes with PBS-0.05% Triton-X. A goat anti-rabbit Alexa647 secondary antibody (Invitrogen, #A21245) and Hoechst 33142 (Invitrogen, #H3570) in blocking solution were then used to incubate cells for one hour at room temperature. Cells were then washed 5 times with PBS-0.05% Triton-X and imaged in PBS.

### Testing rTetR variants in *E. coli*
*E. coli* AB360[50] were co-transfected with rTetR variants and tetO-mCherry reporter plasmid and sequence-verified. For experiments, cultures were first grown overnight at 37 °C with shaking at 230 rpm in sterile-filtered M9 medium (ThermoFisher, #A1374401) supplemented with 0.2% tryptone (MP Biomedicals, #91010817), 0.4% glucose (Sigma-Aldrich, #G8270), 0.001% thiamine (Sigma-Aldrich, #T1270), 0.00006% ferric citrate (Sigma-Aldrich, #F3388), 0.1 mM calcium chloride (Sigma-Aldrich, #C7902), 1 mM magnesium sulfate (Sigma-Aldrich, #M2643) and dox at 50 ng/ml (to reduce accumulation of mCherry). These cultures were diluted 1:20,000 into fresh M9 medium containing the different concentrations of dox immediately prior to the experiment. After 4 h, bacteria were fixed with fixation buffer (Abcam, #AB314680) and mCherry expression was quantified with the BD LSR II cell analyzer (BD Biosciences, 14,000 events were recorded). Data represents the Mean Fluorescence Intensity (MFI) normalized to the vehicle condition.

### Embryoid bodies (EBs)
Induced pluripotent stem cells (iPSC; Stem Cell Technologies, #SCTi003-A) derived from healthy female donor peripheral blood mononuclear cells were maintained using defined MTSER™1 media (Stem Cell Technologies, #85850) in feeder-free conditions using Matrigel-coated plates (Corning, #354277). Cells were passaged using ReLeSR (Stem Cell Technologies, #100-0483) unless experiments required single cells for plating. Single cell plating was performed using Accutase (Stem Cell Technologies, #07922) followed by media supplemented with 10 µM ROCK inhibitor (Y-27632) (Stem Cell Technologies, #72307) for < 24 h. Stem cells were transduced with lentivirus containing TetOn-3G or TetOn-3G_M59I/D5N (Supplementary Data 6). Embryoid bodies (EBs) were generated as described in ref. 51. Briefly, on day 0 of differentiation, 9000 cells per EB were plated using 100 µL of EB Formation Medium (Stem Cell Technologies, #08570) onto ultra-low attachment µ-bottom plates. Medium was changed every other day until 5 days in vitro (DIV5) when medium was changed to induction medium (Stem Cell Technologies, #08570). For all studies, respective concentrations of dox were administered 24 h prior to imaging. Imaging was performed on DIV8.

### Epifluorescence microscopy
Phase contrast and eGFP fluorescence images were obtained on an Axio Vert.A1 FL (Zeiss) microscope fitted with an AxioCam ICM1 camera (Zeiss 60N-C 2/3" 0.63X adapter) at 5X magnification. eGFP images were captured with a BP475/40 excitation and BP530/50 emission filter (FT500 beam splitter). Images were collected with Zen 2 Blue Edition (Zeiss, version 2.0.0.0).

## High-throughput imaging

**Fixed.** Cells were imaged using the Opera Phenix Plus High-Content Screening system. EBs were live-stained with DRAQ5 (Thermo Scientific, #62251) for 20 mins before imaging total cumulative fluorescence on DIV8. High-throughput image analysis was performed using Harmony Software V5.1 (Perkin Elmer).

**Live.** Cells were imaged using the Opera Phenix Plus High-Content Screening system. Live cell tracking was performed by detecting nuclear fluorescence across a period of 24 h (~11 min intervals) using Harmony Software V5.1 (Perkin Elmer). Tracked objects with detectable nuclear GFP signal were exported.

## Protein structure prediction

The AlphaFold2_mmseqs2 Google Colab notebook from ColabFold (v1.5.2-patch; https://colab.research.google.com/github/sokrypton/ColabFold/blob/main/AlphaFold2.ipynb; https://github.com/sokrypton/ColabFold) was used to predict protein structures using default settings. The top-ranked prediction by average pLDDT was used for annotation and visualization with UCSF Chimera (Version 1.17.3), developed by the Resource for Biocomputing, Visualization, and Informatics at the University of California, San Francisco, with support from NIH P41-GM103311[52].

## Statistics

Statistical analyzes were performed in GraphPad Prism 9.2.0, GraphPad Software, San Diego, California USA, www.graphpad.com. All data were collected from biologically distinct samples and plotted as mean ± SEM. VLV titer plots and AF plots were analyzed with two-tailed unpaired t-tests. Cisplatin treatment assays were analyzed with two-tailed t-tests (unpaired in Supplementary Fig. 7b and paired in Supplementary Fig. 7c). Fold-changes in luciferase activity were statistically analyzed with a repeated measures one-way ANOVA with the Geisser-Greenhouse correction with a Dunnett's multiple comparisons test (Fig. 4b), a repeated measures one-way ANOVA with a Dunnett's multiple comparisons test (Fig. 4f) or a Kruskal-Wallis test with Dunn's multiple comparisons test (Supplementary Fig. 6e). Means were compared to the control baseline mean. Changes in nuclear eGFP over time were analyzed using a two-way ANOVA with Šídák's multiple comparisons test (Fig. 3g). RNA editing for ADAR knockout cells was analyzed using an ordinary one-way ANOVA with Dunnett's multiple comparisons test. $P$ values < 0.05 were considered significant.

## Reporting summary

Further information on research design is available in the Nature Portfolio Reporting Summary linked to this article.

# Data availability

The authors confirm that all data supporting findings of this study are available within the article, the Supplementary Information and the Source Data. Source data are provided with this paper. Basecalled long-read nanopore sequencing FASTQ reads and raw short-read Illumina sequencing FASTQ reads have been deposited at the Gene Expression Omnibus (GEO; GSE250502). PROTEUS plasmids will be made available for academic use. Source data are provided with this paper.

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

## Acknowledgements

The authors wish to acknowledge funding from the National Health and Medical Research Council (GNT2029754 to DH, AJC; GNT1185002 to DH, GGN; GNT1107514 to GGN; GNT1158164 to GGN; and GNT1158165 to GGN), the Tour de Cure Mid-Career Research Grant 2023 (G219180 to AJC), The Centenary Institute (DH), the NSW Ministry of Health (DH, AJC, GGN) and a kind donation from Dr. John and Anne Chong (GGN). We also thank Sydney Cytometry for their help with the BD LSR II cell analyzer and Manuela Florido for help with molnupiravir treatments. Schematics were generated using BioRender.com as indicated in figure legends.

## Author contributions

Conceptualization: A.J.C., C.E.D., D.H., G.G.N., Methodology: A.J.C., C.E.D., L.H., D.H., G.G.N., Investigation: A.J.C., C.E.D., C.L.M., L.H., T.D., Visualization: A.J.C., C.E.D., D.H., G.G.N., Funding acquisition: D.H., G.G.N., Project administration: D.H., G.G.N., Supervision: D.H., G.G.N., Writing – original draft: A.J.C., C.E.D., D.H., G.G.N., Writing – review & editing: A.J.C., C.E.D., L.H., D.H., G.G.N.

## Competing interests

A.J.C., C.E.D., D.H., and G.G.N. have filed a provisional patent application on this technology (Australian Patent #2023904160). The remaining authors declare no competing interests.

## Inclusion and Ethics

Our study has included local researchers through all aspects of the research process. This work is globally relevant to researchers in the field. Roles and responsibilities were agreed upon over the course of the research. No severe restrictions prohibited our work. Our work does not require ethics approval but appropriate biosafety approval was sought and provided. Risk to the researchers was minimized through appropriate training protocols, safe work practices and the wearing of PPE. We have not taken local and regional research into account when determining citations.
