## [Transparent Peer Review file · Nature Communications]

A chimeric viral platform for directed evolution in mammalian cells

Corresponding Author: Professor G Neely

Version 0:

Reviewer comments:

Reviewer #1

(Remarks to the Author)

The authors have addressed my earlier comments with their revisions and their work is strengthened, particularly with their clever use of molnupiravir. In their final manuscript they may wish to briefly mention the anticipated cost of adding this reagent during directed evolution campaigns.

Reviewer #2

(Remarks to the Author)

The authors have appropriately addressed my comments. I congratulate them on a very nice piece of work, and am excited to watch the potential of this platform play out in the years to come.

We would like to thank the two reviewers for their valuable time and feedback on our transferred manuscript “A chimeric viral platform for directed evolution in mammalian cells”.

Please find below our responses to each of the comments raised during the review process. We are grateful for the opportunity to improve our manuscript for publication in *Nature Communications*.

Reviewer #1

Remarks to the Author: *The authors have addressed my earlier comments with their revisions and their work is strengthened, particularly with their clever use of molnupiravir. In their final manuscript they may wish to briefly mention the anticipated cost of adding this reagent during directed evolution campaigns.*

RESPONSE: We appreciate the reviewer’s positive response to our improved manuscript and thank them for their review. Molnupiravir is a relatively cheap additive, with approximate costs of ~\$350 USD to supplement 4 litres of culture volume. In 6-well plates, with between 1-2 mL of culture medium used per well, this could cover 2000-4000 evolution samples. Because of the low cost of this addition, we did not feel it was necessary to mention this within the final manuscript.

Reviewer #2

Remarks to the Author: *The authors have appropriately addressed my comments. I congratulate them on a very nice piece of work, and am excited to watch the potential of this platform play out in the years to come.*

RESPONSE: We appreciate the reviewer’s positive response to our improved manuscript and thank them for their review.